# Impact of Cefotaxime Non-susceptibility on the Clinical Outcomes of Bacteremic Pneumococcal Pneumonia

**DOI:** 10.3390/jcm8081150

**Published:** 2019-08-01

**Authors:** Catia Cillóniz, Cristina de la Calle, Cristina Dominedò, Carolina García-Vidal, Celia Cardozo, Albert Gabarrús, Francesc Marco, Antoni Torres, Alex Soriano

**Affiliations:** 1Department of Pneumology, Institut Clinic del Tórax, Hospital Clinic of Barcelona-Institut d’Investigacions Biomèdiques August Pi i Sunyer (IDIBAPS), University of Barcelona (UB)-SGR 911- Ciber de Enfermedades Respiratorias (Ciberes), 08036 Barcelona, Spain; 2Department of Infectious Disease, Hospital Clinic of Barcelona, 08036 Barcelona, Spain; 3Department of Anesthesiology and Intensive Care Medicine, Fondazione Policlinico Universitario A. Gemelli, Università Cattolica del Sacro Cuore, 00168 Rome, Italy; 4Department of Microbiology, Hospital Clinic of Barcelona, 08036 Barcelona, Spain

**Keywords:** cefotaxime, pneumonia, bacteremia, outcomes, infectious

## Abstract

Background: We aimed to analyze the impact of cefotaxime non-susceptibility on the 30-day mortality rate in patients receiving a third-generation cephalosporin for pneumococcal bacteremic pneumonia. Methods: We conducted a retrospective observational study of prospectively collected data from the Hospital Clinic of Barcelona. All adult patients with monomicrobial bacteremic pneumonia due to *Streptococcus pneumoniae* and treated with a third-generation cephalosporin from January 1991 to December 2016 were included. Risk factors associated with 30-day mortality were evaluated by univariate and multivariate analyses. Results: During the study period, 721 eligible episodes were identified, and data on the susceptibility to cefotaxime was obtainable for 690 episodes. Sixty six (10%) cases were due to a cefotaxime non-susceptible strain with a 30-day mortality rate of 8%. Variables associated with 30-day mortality were age, chronic liver disease, septic shock, and the McCabe score. Infection by a cefotaxime non-susceptible *S. pneumoniae* did not increase the mortality rate. Conclusion: Despite the prevalence of cefotaxime, non-susceptible *S. pneumoniae* has increased in recent years. We found no evidence to suggest that patients hospitalized with bacteremic pneumonia due to these strains had worse clinical outcomes than patients with susceptible strains.

## 1. Introduction

*Streptococcus pneumoniae* (pneumococcus) is considered the main cause of non-invasive and invasive infectious diseases that are associated with high morbidity and case fatality rate worldwide, especially in vulnerable populations, such as children and adults aged > 65 years [1,2,3]. The case fatality rates of the invasive pneumococcal disease are 10–30% in adults and <3% in children [4]. The introduction of the 7-valent and 13-valent pneumococcal conjugate vaccines (PCV7, PCV13) in the children population has had a significant impact on the occurrence of pneumococcal infections in children and adults. Several studies reported a reduction in the proportion of invasive pneumococcal disease caused by antibiotic resistant pneumococcal strains; however during recent years we have observed serotype replacement, emergence of non-vaccine serotypes, and infections caused by non-encapsulated pneumococcus [5,6,7]. In Catalonia, where this study was carried out, the coverage of PCV7 was 48% in 2007 in children, whereas in 2015 the global vaccination coverage in adults was 39% (69% for 23-valent pneumococcal polysaccharide vaccine (PPV23) and 0.2% for PCV13). It is important to note that before July 2016 neither PCV7 nor PCV13 was included in the routine vaccination schedule in Catalonia for children, except for those under 5 years old with specific risk factors. In adults, PCV13 was available from 2012 and the reported coverage in 2015 was 0.2% [8,9].

Antimicrobial resistance of pneumococcus is still a global concern [10]; it modifies the clinical presentation of pneumococcal disease, making its diagnosis and treatment more complex, and affecting clinical outcomes [6,11]. Despite the antimicrobial resistance of pneumococcus strains, cefotaxime and ceftriaxone remain the most active cephalosporins [12]. To date, however, the clinical impact of third-generation cephalosporins (3GC) non-susceptibility on clinical outcomes has not been comprehensively evaluated in cases of pneumococcal bacteremic pneumonia.

In the present study, we aimed to analyze the effect of 3GC non-susceptibility on the outcomes of patients with bacteremic pneumococcal pneumonia treated empirically with a third-generation cephalosporin. 

## 2. Methods

### 2.1. Ethics Statement

The study was approved by the Ethics Committee of our institution (Comité Ètic d’Investigació Clínica, register: 2009/5451). The need for written informed consent was waived because of the non-interventional study design.

### 2.2. Study Design and Patients

This was a retrospective observational study of data that were prospectively collected at the Hospital Clinic of Barcelona. All adult patients with monomicrobial bacteremic pneumonia due to *S. pneumoniae* admitted to the hospital between January 1991 and December 2016 were included (*n* = 2490). We excluded cases of bacteremic pneumonia not caused by *S. pneumoniae* and cases with incomplete clinical data. All included patients received a third-generation cephalosporin empirically either as monotherapy, or in combination with a macrolide, or a fluoroquinolone. The patients were then divided into two groups based on their susceptibility to 3GC. 

### 2.3. Data Collection and Evaluation

The following data were obtained from all patients: Age, gender, comorbidities, McCabe score [13], Pneumonia Severity Score (PSI) [14] was calculated in 312 cases, recent antimicrobial or steroid treatment (within the last month), recent hospitalization (within the last month), surgery and other invasive procedures, shock at presentation, etiology (microorganisms), empirical antimicrobial treatment, appropriateness of empirical therapy, and 30-day mortality. Patients were then prospectively followed up by a senior infectious disease specialist who assessed the medical history, physical examinations, microbiological tests, and complementary imaging to determine the source of infection.

### 2.4. Definitions

Pneumonia was defined as the presence of a new pulmonary infiltrate on chest x-ray at a hospital admission with symptoms and signs of lower respiratory tract infection. Pneumococcal bacteremia was diagnosed when a pneumococcal isolate was recovered from a blood culture of patients with pneumonia. Underlying diseases were classified according to the modified McCabe and Jackson criteria into rapidly fatal, finally fatal, or non-fatal [13].

Prior antimicrobial therapy was defined as the use of any antimicrobial agent for at least three days during the month before the onset of bacteremia. Prior steroid therapy was defined as the use of at least 10 mg of prednisone (or equivalent dose of another steroid) during the month before admission. Septic shock was defined as a systolic blood pressure < 90 mmHg, peripheral hypoperfusion, and use of vasopressors for > 4 h after initial fluid replacement [15].The most widely used third-generation cephalosporin in our institution (>90%) is ceftriaxone, the standard dose is 1 g q24h for non-critically ill patients and 2 g q24h for critically ill patients.

### 2.5. Microbiological Evaluation and Diagnostic Criteria

During the study period, blood cultures were processed using a BACTEC 9240 system (Becton eDickinson Microbiology Systems), with an incubation period of 5 days. Isolates were identified by standard techniques. The minimum inhibitory concentrations (MIC) of *S. pneumoniae* isolates were determined using the E-test method and broth microdilution (Sensititre, Trek Diagnostic Systems, West Sussex, UK), for penicillin, cefotaxime, ceftriaxone, cefepime, imipenem, meropenem, erythromycin, clindamycin, levofloxacin, and vancomycin. Ceftriaxone susceptibility results were categorized as susceptible (MIC ≤ 0.5 μg/mL) and non-susceptible (intermediate, MIC > 0.5 to ≤ 2 μg/mL and resistant, MIC > 2 µg/mL). 

### 2.6. Statistical Analysis

We report the number and percentage of patients for categorical variables, and the mean and standard deviation (SD) for continuous variables. Categorical variables were compared using the χ^2^ test or the Fisher exact test, whereas continuous variables were compared using the *t*-test. Logistic regression analyses [16,17] were then used to examine the associations between 30-day mortality and risk factors. First, each risk factor was tested individually. Second, all risk factors that showed an association in the univariate model (*p* < 0.10) were added to the multivariate model. Finally, a backward stepwise selection (*p*_in_ < 0.05, *p*_out_ > 0.10) was used to determine the factors associated with 30-day mortality. Multicollinearity was assessed using the variance inflation factor. Odds ratios (ORs) and 95% confidence intervals (CIs) were calculated. The Hosmer–Lemeshow goodness-of-fit test was performed to assess the overall fit of the multivariable model. We also calculated the area under the receiver operating characteristic curve (AUC) of the final model. To evaluate possible overfitting and instability of the selection variables, we performed internal validation using ordinary nonparametric bootstrapping with 1000 samples and bias-corrected, accelerated 95% CIs [18]. All analyses were performed with IBM SPSS, Version 23.0 (IBM Corp., Armonk, NY, USA), and the significance level was set at 0.05 (two-tailed).

## 3. Results

### 3.1. Patients’ Characteristics

During the study period, we identified 721 episodes of monomicrobial bacteremic pneumococcal pneumonia; of these, 31 were excluded because they were missing important data. Finally, in 690 episodes of pneumococcal bacteremia were included. The prevalence of 3GC non-susceptible *S. pneumoniae* strains during the study is shown in Figure 1.

Our cohort comprised 439 males (64%) and 251 females (36%), with a mean (SD) age of 66.5 (19.9) years; notably, 309 (45%) were aged > 65 years. The main comorbidities were chronic obstructive pulmonary disease (21%), human immunodeficiency virus infection (20%), diabetes mellitus (16%), and chronic liver disease (9%). Overall, 94 cases (14%) presented with septic shock, and the 30-day mortality was 47 (7.0%). One hundred eighty-three (27%) patients received a third-generation cephalosporin empirically as monotherapy, or in combination with a macrolide (*n* = 350; 51%), or a fluoroquinolone (*n* = 157; 23%).

We identified that 66 cases (10%) were due to a 3GC non-susceptible *S. pneumoniae*. Notably, the prevalence of these strains did not change significantly over time (*p* = 0.75) (Figure 2). 

The characteristics of patients according to 3GC susceptibility are shown in Table 1. Patients with 3GC non-susceptibility were almost 10 years older than those with a susceptible strain (*p* = 0.003). Most cases had a pneumonia severity index of IV–V (68%) and a McCabe score of 1 (66%).

### 3.2. Microbiology

Data regarding serotypes were available in 134 cases; serotypes covered by the PCV7 vaccine (4, 6B, 9V, 14, 18C, 19F, and 23F) and PCV13 (1, 3, 4, 5, 6A, 6B, 7F, 9V, 14, 18C, 19A, 19F and 23F) represented 13% and 68% of all serotypes, respectively. The most frequent serotypes in the susceptible group were: serotype 1 (25 cases, 20%), 19A (15 cases, 12%), 3 (15 cases, 12%) and 7F (11 cases, 9%). Whereas, three serotypes (8 cases, 6%) were reported in the non-susceptible group: 11A (2 cases), 14 (5 cases), and 19F (1 case). The empiric antimicrobial therapy given was comparable between the susceptible and non-susceptible groups. 

### 3.3. Clinical Outcomes

We found no evidence that patients with 3GC non-susceptible *S. pneumoniae* bacteremic pneumonia presented with more severe disease or had worse clinical outcomes (Table 1).

### 3.4. Factors Associated with 30-Day Mortality

The univariate logistic regression analysis revealed several variables significantly associated with 30-day mortality (Table 2). Among these, age, presence of septic shock at admission, chronic liver disease, and McCabe score remained independently associated with 30-day mortality in the multivariate analysis. 

The AUC was 0.85 (95% CI, 0.78–0.92) for the 30-day mortality in this model. Internal validation of the logistic regression model (bootstrapping with 1000 samples) demonstrated robust results for all variables included in the model, with small 95% CIs around the original coefficients.

## 4. Discussion

The main findings of our study are as follows. First, although we observed an increase in the prevalence of 3GC non-susceptible strains of *S. pneumoniae* in recent years, this prevalence did not change significantly over time. Second, we found no evidence of more severe presentations or worse clinical outcomes in patients admitted to the hospital with 3GC non-susceptible *S. pneumoniae* bacteremic pneumonia. Third, the 30-day mortality was shown to be independently associated with age, septic shock at admission, chronic liver disease, and the McCabe score.

*S. pneumoniae* remains the leading cause of non-invasive and invasive disease worldwide [19]. International guidelines for the management of pneumococcal pneumonia recommend the use of β-lactams, such as a 3GC (cefotaxime and ceftriaxone) [20,21]. Although resistance to 3GC remains low worldwide, an increase in the rate of resistance has been reported in many countries due to the spread of pneumococcal resistant clones [22,23]. Although we reported that the prevalence of 3GC non-susceptibility in pneumococcal bacteremic pneumonia did not change significantly over time during the study period, we observed an increase in the prevalence from 0% in the first period (1991–1995) to 13% in the last period (2011–2016). This observation is consistent with several reports about the global emergence of in vitro antimicrobial resistance in *S. pneumoniae* [24,25,26,27]. Interestingly, in the non-susceptible group, the majority of the cases were caused by serotypes covered by PCV13 (75%) and PCV7 (63%) vaccines. Other possible explanations could be replacement because of a lack of vaccine protection. However, the vaccination status of our study population is unknown, but two recent studies reported low-intermediate pneumococcal vaccination coverage in our region [8,9]; this could explain the temporal variation that we observed in our study. Unfortunately, without data about vaccination status we cannot make more conclusions. 

Information about the impact of 3GC non-susceptible *S. pneumoniae* strains on the outcomes of patients with bacteremic pneumonia is scarce. In 2001, Moroney et al. [28] performed a case-control study and found that drug resistance (cefotaxime non-susceptible) did not demonstrably affect mortality or intensive care unit (ICU) admissions among patients hospitalized with bacteremic or other forms of invasive pneumococcal pneumonia. In 2012, Song et al. [29] published an 11-year study evaluating the risk factors for mortality, and reported the impact of antimicrobial resistance (erythromycin and penicillin resistance) on the clinical outcomes of patients with pneumococcal bacteremia. Whereas an elevated APACHE II score and the presence of solid organ tumors were independently associated with mortality, neither erythromycin resistance nor penicillin resistance significantly affected clinical outcomes. One of the few studies that investigates ceftriaxone non-susceptibility was the study by Choi et al. [30] that investigated the impact of penicillin non-susceptibility on the clinical outcomes of patients with non-meningeal pneumococcal bacteremia and reported that the 30-day mortality was similar between patients with resistant and susceptible strains. In the multivariate analysis of this study, ceftriaxone non-susceptibility (OR 4.88; 95% CI 1.07–22.27; *p* = 0.041) was an independent risk factor for 30-day mortality, but not all patients were treated with a 3GC.

For the present study, we applied the 3GC breakpoints set out in the 2008 Clinical and Laboratory Standards Institute guidance. All the patients included in the study received a 3GC as empirical treatment. The independent risk factors for 30-day mortality in patients with pneumococcal bacteremic pneumonia in the present study were age, septic shock at admission, chronic liver disease, and McCabe score, but not the MIC of ceftriaxone. The PK/PD parameter that predicts the outcome of ceftriaxone is the time that the serum concentration of ceftriaxone is above the MIC (T > MIC). According to prior data, at least 40% of the interval between two consecutive doses is the needed exposure to obtain a bacteriostatic effect [31]. Giving 1 or 2 g q24 h, the probability of attaining this target when the MIC of ceftriaxone is 1 µg/mL (intermediate susceptibility) is high and this explains our results. However, the number of strains resistant to ceftriaxone (MIC > 2 µg/mL) were few and we cannot predict the response with our cohort.

Previous studies have also emphasized the importance of host factors when predicting severity and outcomes. To date, older age, alcohol abuse, nursing-home residence [32], ICU admission, platelet count < 100,000/μL [33], and liver disease [34] have each been significantly associated with mortality. These data support the idea that host factors, mainly chronic comorbidities, are related to disease severity and clinical outcomes.

Our study has some limitations. First, the length of the recruitment period (27 years), has undoubtedly been associated with major advances in patient care. However, we have followed the 1993, 1998, 2001, and 2007 guidelines for managing CAP [20,35,36,37]. Additionally, our management protocol for pneumonia did not change substantially. Second, this study was conducted at a single centre, which necessitates cautious extrapolation of the findings to other settings. Third, the number of patients with non-susceptible strains was low and the majority of the patients received combination treatment with a macrolide or a fluoroquinolone. The influence of ceftriaxone alone could not be evaluated. However, the current guidelines [20] recommend combination therapy with a macrolide and so our study shows that non-susceptibility to ceftriaxone does not affect the efficacy of the recommended regimen. Fourth, complete information to calculated PSI scores was not collected in the first years of study. Fifth, there was a lack of data about vaccination status. However, two previous studies reported that Catalonia, the region were this study was carried out, has a low-intermediate pneumococcal vaccine coverage [8,9]. 

## 5. Conclusions

In conclusion, we found no evidence to suggest that patients hospitalized for 3GC non-susceptible *S. pneumoniae* bacteremic pneumonia had worse clinical outcomes than patients admitted with antimicrobial-susceptible pathogens. 

## Figures and Tables

**Figure 1 jcm-08-01150-f001:**
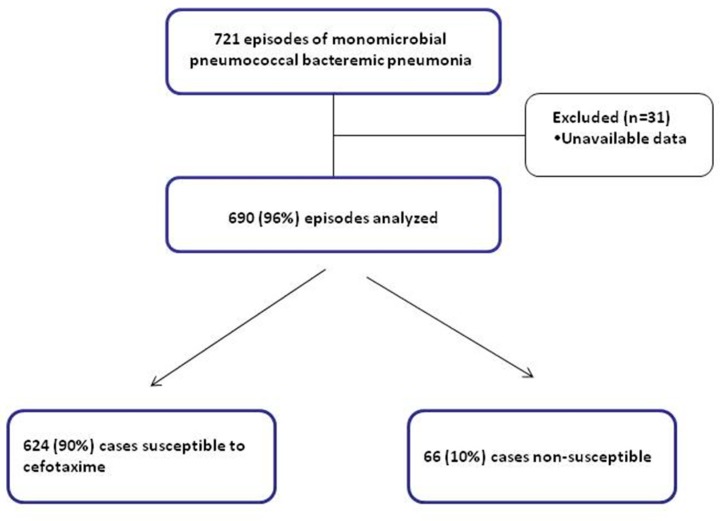
Flow diagram of population selection.

**Figure 2 jcm-08-01150-f002:**
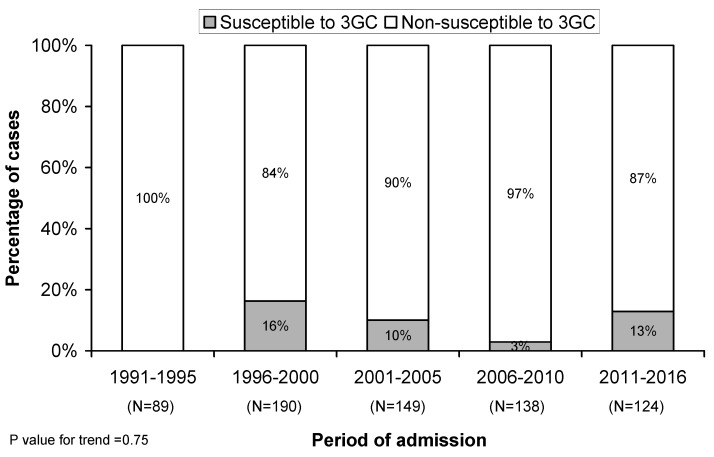
Temporal distribution of susceptible and non-susceptible cefotaxime during the study.

**Table 1 jcm-08-01150-t001:** Patient demographics and clinical characteristics at admission.

	Non-Susceptible to 3GC (*N* = 66)	Susceptible to 3GC (*N* = 624)	*p*-Value
Age, years, mean (SD)	67 (20)	59 (19)	0.001
Age > 65 years, *n* (%)	42 (64)	267 (43)	0.001
Male, gender, *n* (%)	38 (58)	401 (64)	0.28
Current alcohol user, *n* (%)	4 (6)	34 (5)	0.78
Previous antimicrobials, *n* (%)	4 (6)	19 (3)	0.26
Shock, *n* (%)	8 (12)	85 (14)	0.73
Previous systemic steroids, *n* (%)	4 (6)	50 (8)	0.54
Mechanical ventilation, *n* (%)	0	4 (1)	>0.99
Fever, *n* (%)	63 (95)	605 (97)	0.42
Comorbidities, *n* (%)			
COPD	17 (26)	128 (21)	0.32
HIV	14 (21)	122 (20)	0.75
Neoplasm	14 (21)	98 (16)	0.25
Chronic cardiovascular disease	9 (14)	59 (9)	0.28
Diabetes mellitus	15 (23)	96 (15)	0.12
Chronic renal disease	3 (5)	30 (5)	>0.99
Chronic liver disease	10 (15)	74 (12)	0.44
Empiric therapy			
Cephalosporin monotherapy	19 (29)	164 (26)	
Cephalosporin + quinolone Cephalosporin + macrolide	10 (15) 37 (56)	147 (24) 313 (50)	0.16 0.12
McCabe score			0.73
1	43 (66)	438 (71)	
2	21 (32)	174 (28)	
3	1 (2)	7 (1)	
PSI score			0.11
PSI I–III	8 (32)	140 (49)	
PSI IV–V	17 (68)	147 (51)	
30-day mortality, *n* (%)	5 (8)	42 (7)	0.80

Data are number of patients (%) or mean (standard deviation). Percentages were calculated on non-missing data. Abbreviations: COPD, chronic obstructive pulmonary disease; HIV, human immunodeficiency virus; PSI, pneumonia severity index.

**Table 2 jcm-08-01150-t002:** Univariate and multivariate logistic regression analyses of predictors of 30-day mortality.

Variable	Univariate ^a^			Multivariable ^b,c^		
	OR	95% CI	*p*-Value	OR	95% CI	*p*-Value
Non-susceptible to 3GC	1.13	0.43–2.96	0.81	0.83	0.28–2.50	0.74
Age (+1 year)	1.03	1.01–1.05	0.001	1.04	1.02–1.06	<0.001
Septic shock	13.01	6.89–24.59	<0.001	17.54	8.72–35.27	<0.001
Chronic renal disease	3.89	1.05–14.46	0.042	-	-	-
Chronic liver disease	2.37	1.16–4.86	0.019	2.33	0.94–5.75	0.068
McCabe score ^d^			0.009			0.061
1	1.00	-	-	1.00	-	-
2	1.45	0.76–2.74	0.26	1.32	0.60–2.89	0.49
3	9.66	2.20–42.51	0.003	8.72	1.43–53.00	0.019

Abbreviations: CI, confidence interval; OR, odds ratio. Data are shown as estimated ORs (95% CIs) of the explanatory variables in the 30-day mortality group. The OR is defined as the probability of membership of the group, 30-day mortality, divided by the probability of membership of the non-30-day mortality group. The *p*-value is based on the null hypothesis that all ORs relating to an explanatory variable equal unity (no effect). The variables analyzed in the univariate analysis were: age, gender, McCabe score, cefotaxime, cephalosporin macrolide empiric therapy, cephalosporin quinolone empiric therapy, cephalosporin empiric, quinolone empiric, empiric, macrolide empiric, community origin, fiber, shock, corticosteroid therapy, diabetes, respiratory disease, hepatic diseases, alcohol, COPD, HIV, chronic renal disease, previous antibiotic therapy, acute renal failure. Adjusted for the antimicrobial susceptibility of pneumococcal bacteremia to 3GC. Hosmer-Lemeshow goodness-of-fit test, *p* = 0.28. The *p*-value corresponds to differences between the three groups (1, 2, or 3).

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
