# Peer review of "Impact of Cefotaxime Non-susceptibility on the Clinical Outcomes of Bacteremic Pneumococcal Pneumonia"

_jcm, 2019, doi:10.3390/jcm8081150_

Round 1

Reviewer 1 Report

Thank you for your submission. The topic you are addressing in this study is an important one. The methods and results are logically laid out. The discussion about why or why not resistance might affect the outcomes studied is clear. A few suggestions that might clarify and strengthen the arguments are:

There are several large studies that conclude that the introduction of pneumococcal conjugate vaccines have reduced pneumococcal antimicrobial resistance. One reference for this would be Kim L, et al. Clin Microbiol Rev 2016. I recommend raising this issue in the background. It seems you might have observed the same through 2010, but then there was an increase in resistance from 2011-2016 (Figure 2). Addressing possible reasons for the increase in antimicrobial resistance in 2011-2016 seems necessary. Do you think there is replacement occurring or a lack of vaccine protection (lines 179-189)? Even if the vaccination status of cases is unknown, it would be helpful to report the coverage rates by age strata to help understand the results.

Figure 2: is this the percentage of all cases or all cases tested for resistance?

Line 169: is the point here that the total number of resistance cases didn’t change over time, even though the proportion increased in the 2011-2016 time period? I think I might be missing point here, could it be elaborated on?

Did any of the lab techniques described in lines 88-93 change over the study period?

Why do you think there is such a difference the proportion of males vs females?

Minor point: I think the “30-day mortality rate” reported is actually a case fatality ratio, right?

The response to some of the above questions might help to expand what currently seems an incomplete limitations section. Lastly, in the conclusions you might want add a sentence to highlight the reasons why there was no difference in clinical outcomes observed.

All the best.

Author Response

Reviewer 1: Thank you for your submission. The topic you are addressing in this study is an important one. The methods and results are logically laid out. The discussion about why or why not resistance might affect the outcomes studied is clear. A few suggestions that might clarify and strengthen the arguments are:

Many thanks to the reviewer for their comments

1. - There are several large studies that conclude that the introduction of pneumococcal conjugate vaccines have reduced pneumococcal antimicrobial resistance. One reference for this would be Kim L, et al. Clin Microbiol Rev 2016. I recommend raising this issue in the background. It seems you might have observed the same through 2010, but then there was an increase in resistance from 2011-2016 (Figure 2). Addressing possible reasons for the increase in antimicrobial resistance in 2011-2016 seems necessary. Do you think there is replacement occurring or a lack of vaccine protection (lines 179-189)? Even if the vaccination status of cases is unknown, it would be helpful to report the coverage rates by age strata to help understand the results.

Answer: thanks for the comments. We modified our introduction section (page 1, line 44 to 62) and mentioned the reference that the reviewer recommends (reference 5). We also reported the vaccination coverage in Catalonia the region where this study was carrying out (page 2, line 53 to 59). In discussion section (page 6, line 190 to 196) we included a paragraph about the possible explanation to our observation of the temporal changes of non-susceptible cefotaxime.

2. - Figure 2: is this the percentage of all cases or all cases tested for resistance?

Answer: thanks for the comment. Our study population comprised all cases tested for resistance (n=690). Data on figure 2 showed the percentage of non-susceptible cefotaxime strains during the period of study.

Period 1991-95: 0/89

Period 1996-00: 31/190

Period 2001-05: 15/149

Period 2006-10: 4/138

Period 2011-16: 16/124

Total study: 66/690

3.-Line 169: is the point here that the total number of resistance cases didn’t change over time, even though the proportion increased in the 2011-2016 time period? I think I might be missing point here, could it be elaborated on?

Answer: The p values for trend were not statistically significant for both the period of admission (0.746) and year of admission (p=0.882):

Crosstabulation

Non-susceptible to 3GC

Susceptible to 3GC

Total

N

%

N

%

N

%

Period   of admission

1991-1995

0

0,0%

89

100,0%

89

100,0%

1996-2000

31

16,3%

159

83,7%

190

100,0%

2001-2005

15

10,1%

134

89,9%

149

100,0%

2006-2010

4

2,9%

134

97,1%

138

100,0%

2011-2016

16

12,9%

108

87,1%

124

100,0%

Total

66

9,6%

624

90,4%

690

100,0%

Chi-Square Tests

Value

df

Asymptotic Significance   (2-sided)

Pearson   Chi-Square

28,154a

4

,000

Likelihood   Ratio

37,343

4

,000

Linear-by-Linear   Association

,105

1

,746

N of   Valid Cases

690

a. 0 cells (0,0%) have expected count less than 5.   The minimum expected count is 8,51.

Crosstabulation

Non-susceptible to 3GC

Susceptible to 3GC

Total

N

%

N

%

N

%

Year   of admission

1991

0

0,0%

12

100,0%

12

100,0%

1992

0

0,0%

20

100,0%

20

100,0%

1993

0

0,0%

13

100,0%

13

100,0%

1994

0

0,0%

18

100,0%

18

100,0%

1995

0

0,0%

26

100,0%

26

100,0%

1996

5

17,9%

23

82,1%

28

100,0%

1997

7

20,6%

27

79,4%

34

100,0%

1998

6

14,3%

36

85,7%

42

100,0%

1999

8

16,0%

42

84,0%

50

100,0%

2000,

5

13,9%

31

86,1%

36

100,0%

2001

5

14,3%

30

85,7%

35

100,0%

2002

3

9,1%

30

90,9%

33

100,0%

2003

6

20,7%

23

79,3%

29

100,0%

2004

0

0,0%

17

100,0%

17

100,0%

2005

1

2,9%

34

97,1%

35

100,0%

2006

0

0,0%

23

100,0%

23

100,0%

2007

1

2,5%

39

97,5%

40

100,0%

2008

0

0,0%

31

100,0%

31

100,0%

2009

1

4,8%

20

95,2%

21

100,0%

2010

2

8,7%

21

91,3%

23

100,0%

2011

2

11,1%

16

88,9%

18

100,0%

2012

3

10,7%

25

89,3%

28

100,0%

2013

5

20,8%

19

79,2%

24

100,0%

2014

3

15,0%

17

85,0%

20

100,0%

2015

2

9,5%

19

90,5%

21

100,0%

2016

1

7,7%

12

92,3%

13

100,0%

Total

66

9,6%

624

90,4%

690

100,0%

Chi-Square Tests

Value

df

Asymptotic Significance   (2-sided)

Pearson   Chi-Square

42,297a

25

,017

Likelihood   Ratio

55,199

25

,000

Linear-by-Linear   Association

,022

1

,882

N of   Valid Cases

690

a. 26 cells   (50,0%) have expected count less than 5. The minimum expected count is 1,15.

4. - Did any of the lab techniques described in lines 88-93 change over the study period?

Answer: thanks for the comment. There were not changes over time in the techniques described in the microbiological diagnostic section. The use of the E-test is occasional, only in case of doubt or to confirm a result.

5. - Why do you think there is such a difference the proportion of males vs females?

Answer: We did not find differences in the proportion regard gender in our study: in table 1 the proportions were: males 58% vs. female 64%; p 0.28. However it is well known that males have increasing risk of pneumonia because of the higher proportion of previous comorbidities[1].

                1.            Torres, A.; Peetermans, W.E.; Viegi, G.; Blasi, F. Risk factors for community-acquired pneumonia in adults in Europe: a literature review. Thorax 2013, 68, 1057–1065.

6. - Minor point: I think the “30-day mortality rate” reported is actually a case fatality ratio, right?

Answer: thanks for the comment. Reviewer is right that we can considered “30-day mortality” (the number of patients who died at 30 day post-admission from hospital) as case fatality ratio.

7. - The response to some of the above questions might help to expand what currently seems an incomplete limitations section. Lastly, in the conclusions you might want add a sentence to highlight the reasons why there was no difference in clinical outcomes observed.

Answer: thanks for the comment. We modified our limitation section accordingly to reviewer comments and added the reasons for the differences in clinical outcomes observed in our discussion section (page 8, line 246 to 258).

Reviewer 2 Report

In this manuscript "Impact of Cefotaxime Non-susceptibility on the Clinical Outcomes of Bacteremic Pneumococcal Pneumonia” Catia Cilloniz et.al., in a retrospective observational study, evaluated the effect of third-generation cephalosporin (3GC) non-susceptibility on the outcomes of patients with bacteremic pneumococcal pneumonia.

The comments and suggestions for this manuscript are as follows-

1.      Page 2 line 58-62. Did the author consider the co-infection S. pneumoniae and influenza virus which is very common in the case of pneumonia? What is the percentage of adult patients with monomicrobial bacteremic pneumonia due to S. pneumoniae admitted to the hospital between January 1991 and December 2016 over the total pneumonia cases (coinfection or pneumonia caused by other bacteria)?

2.      Page 2 line 65 and page 3 line 113-114. The author should justify the reason behind the calculation of Pneumonia Severity Score (PSI) in only 312 cases while 690 episodes of pneumococcal bacteremia were included.

3.      Page 4 line 118-124. The author must provide this information in the “2.2 Study design and patients” section.

4.      Page 4 Figure 2. The author may include Temporal distribution of susceptible cefotaxime during the study in the bar graph.

5.      Page 4 table 1, “Patient demographics and clinical characteristics at admission”. (I) Are the clinical characteristics only for male (gender specific) or combined? (II) Are the males more susceptible to the COPD/ pneumonia?

6.      The authors should provide a more comprehensive introduction and discussion for this study.

Author Response

Reviewer 2: In this manuscript "Impact of Cefotaxime Non-susceptibility on the Clinical Outcomes of Bacteremic Pneumococcal Pneumonia” Catia Cilloniz et.al., in a retrospective observational study, evaluated the effect of third-generation cephalosporin (3GC) non-susceptibility on the outcomes of patients with bacteremic pneumococcal pneumonia.

1.      Page 2 line 58-62. Did the author consider the co-infection S. pneumoniae and influenza virus which is very common in the case of pneumonia? What is the percentage of adult patients with monomicrobial bacteremic pneumonia due to S. pneumoniae admitted to the hospital between January 1991 and December 2016 over the total pneumonia cases (coinfection or pneumonia caused by other bacteria)?

Answer: thanks for the comment. We excluded cases of co-infection between S. pneumoniae and other microorganism and included only the monomicrobial cases. During the study period 2490 cases of bacteremic pneumonia were collected, 721 (29%) were monomicrobiana due S. pneumoniae. The rate of polymicrobial (bacterial) infections was 8% (n=200). The information about co-infection with influenza was not available in our database.

2.      Page 2 line 65 and page 3 line 113-114. The author should justify the reason behind the calculation of Pneumonia Severity Score (PSI) in only 312 cases while 690 episodes of pneumococcal bacteremia were included.

Answer: thanks for the comment. Our study comprised data from a large period of study, register of data for calculation PSI score was not completed in the first years of the study, especially the variables of arterial pH and oxygenation. We will include this as a limitation in the new version of the manuscript (page 8, limitations paragraph).

3.      Page 4 line 118-124. The author must provide this information in the “2.2 Study design and patients” section.

Answer: thanks for the comment. We provided the information in page 4 line 118 to 124 in the study design and patients (page 2, line 79- 80).

4.      Page 4 Figure 2. The author may include Temporal distribution of susceptible cefotaxime during the study in the bar graph.

Answer: thanks for the comment. We included a new version of the figure 2 including the temporal distribution of susceptible cefotaxime.

5.      Page 4 table 1, “Patient demographics and clinical characteristics at admission”. (I) Are the clinical characteristics only for male (gender specific) or combined? (II) Are the males more susceptible to the COPD/ pneumonia?

Answer: thanks for the comment. The clinical characteristics in table 1 are for all population (male and female). In our study there were no differences in the proportion of COPD patients between males or females.

Male

(N = 439)

female

(N = 251)

P-value

  COPD

101 (23)

44 (18)

0.089

Data are number of patients (%). Percentages were calculated on non-missing data. Abbreviations: COPD, chronic obstructive pulmonary disease.

6.      The authors should provide a more comprehensive introduction and discussion for this study.

Answer: thanks for the comment. We modified our introduction (page 1) and the discussion (page 7 to 9) in the new version of the manuscript.

Round 2

Reviewer 2 Report

Author response is satisfactory.